# RainNet: A Large-Scale Imagery Dataset for Spatial Precipitation Downscaling

## Abstract

Contemporary deep learning frameworks have been applied to solve meteorological problems (*e.g.*, front detection, synthetic radar generation, precipitation nowcasting, *e.t.c.*) and have achieved highly promising results. Spatial precipitation downscaling is one of the most important meteorological problems. However, the lack of a well-organized and annotated large-scale dataset hinders the training and verification of more effective and advancing deep-learning models for precipitation downscaling. To alleviate these obstacles, we present the first large-scale spatial precipitation downscaling dataset named *RainNet*, which contains more than $62,400$ pairs of high-quality low/high-resolution precipitation maps for over 17 years, ready to help the evolution of deep models in precipitation downscaling. Specifically, the precipitation maps carefully collected in RainNet cover various meteorological phenomena (*e.g.*, hurricane, squall, *e.t.c.*), which is of great help to improve the model generalization ability. In addition, the map pairs in RainNet are organized in the form of image sequences (720 maps per month or 1 map/hour), showing complex physical properties, *e.g.*, temporal misalignment, temporal sparse, and fluid properties. Two machine-learning-oriented metrics are specifically introduced to evaluate or verify the comprehensive performance of the trained model, (*e.g.*, prediction maps reconstruction accuracy). To illustrate the applications of RainNet, 14 state-of-the-art models, including deep models and traditional approaches, are evaluated. To fully explore potential downscaling solutions, we propose an implicit physical estimation framework to learn the above characteristics. Extensive experiments demonstrate that the value of RainNet in training and evaluating downscaling models.

## 1 Introduction

Deep learning has made an enormous breakthrough in the field of computer vision, which is extremely good at extracting valuable knowledge from numerous amounts of data. In recent years, with computer science development, a deluge of Earth system data is continuously being obtained, coming from sensors all over the earth and even in space. These ever-increasing massive amounts of data with different sources and structures challenge the geoscience community, which lacks practical approaches to understand and further utilize the raw data (Reichstein et al. (2019)). Specifically, several preliminary works (Groenke et al. (2020); White et al. (2019); He et al. (2016); Ravuri et al. (2021); Angell & Sheldon (2018); Veillette et al. (2020)) try to introduce machine learning and deep learning frameworks to solve meteorological problems, *e.g.*, spatial precipitation downscaling.

In this paper, we focus on the spatial precipitation downscaling task. Spatial precipitation downscaling is a procedure to infer high-resolution meteorological information from low-resolution variables, which is one of the most important upstream components for meteorological task (Bauer et al. (2015)). The precision of weather and climate prediction is highly dependent on the resolution and reliability of the initial environmental input variables, and spatial precipitation downscaling is the most promising solution. The improvement of the weather/climate forecast and Geo-data quality saves tremendous money and lives; with the fiscal year 2020 budget over $1 billion, NSF funds thousands of colleges in the U.S. to research on these topics (NSF (2020)).

Unfortunately, there are looming issues hinders the research of spatial precipitation downscaling in the machine learning community: 1). Lack of "machine-learning ready" datasets. The existing

machine-learning-based downscaling methods are only applied to ideal retrospective problems and verified on simulated datasets (*e.g.*, mapping bicubic of precipitation generated by weather forecast model to original data (Berrisford et al. (2011))), which significantly weakens the credibility of the feasibility, practicability, and effectiveness of the methods. It is worth mentioning that the data obtained by the simulated degradation methods (*e.g.*, bicubic) is completely different from the real data usually collected by two measurement systems (*e.g.*, satellite and radar) with different precision. The lack of a well-organized and annotated large-scale dataset hinders the training and verification of more effective and complex deep-learning models for precipitation downscaling. 2). Lack of tailored metrics to evaluate machine-learning-based frameworks. Unlike deep learning (DL) and machine learning (ML) communities, scientists in meteorology usually employ maps/charts to assessing downscaling models case by case based on domain knowledge (He et al. (2016); Walton et al. (2020)), which hinders the application of Rainnet in DL/ML communities. For example, (He et al. (2016)) use log-semivariance (spatial metrics for local precipitation), quantile-quantile maps to analyzing the maps. 3). an efficient downscaling deep-learning framework should be established. Contrary to image data, this real precipitation dataset covers various types of real meteorological phenomena (e.g., Hurricane, Squall, *e.t.c.*), and shows the physical characters (*e.g.*, *temporal misalignment*, *temporal sparse* and *fluid properties*, *e.t.c.*) that challenge the downscaling algorithms. Traditional computationally dense physics-driven downscaling methods are powerless to handle the increasing meteorological data size and flexible to multiple data sources.

To alleviate these obstacles, we propose the first large-scale spatial precipitation downscaling dataset named *RainNet*, which contains more than $62,400$ pairs of high-quality low/high-resolution precipitation maps for over 17 years, ready to help the evolution of deep models in spatial precipitation downscaling. The proposed dataset covers more than 9 million square kilometers of land area, which contains both wet and dry seasons and diverse meteorological phenomena. To facilitate DL/ML and other researchers to use RainNet, we introduce 6 most concerning indices to evaluate downscaling models: mesoscale peak precipitation error (MPPE), heavy rain region error (HRRE), cumulative precipitation mean square error (CPMSE), cluster mean distance (CMD), heavy rain transition speed (HRTS) and average miss moving degree (AMMD). In order to further simplify the application of indices, we abstract them into two weighted and summed metrics: Precipitation Error Measure (PEM) and Precipitation Dynamics Error Measure (PDEM). Unlike video super-resolution, the motion of the precipitation region is non-rigid (*i.e.*, fluid), while video super-resolution mainly concerns rigid body motion estimation. To fully explore how to alleviate the mentioned predicament, we propose an implicit dynamics estimation driven downscaling deep learning model. Our model hierarchically aligns adjacent precipitation maps, that is, implicit motion estimation, which is very simple but exhibits highly competitive performance. Based on meteorological science, we also proved that the dataset we constructed contained the full information people may need to recover the higher resolution observations from lower resolution ones.

The main contributions of this paper are:

- To the best of our knowledge, we present the first REAL (non-simulated) Large-Scale Spatial Precipitation Downscaling Dataset for deep learning;

- We introduce 2 simple metrics to evaluate the downscaling models;

- We propose a downscaling model with strong competitiveness. We evaluate 14 competitive potential solutions on the proposed dataset, and analyze the feasibility and effectiveness of these solutions.

## 2 BACKGROUND

At the beginning of the $19^{th}$ century, geoscientists recognized that predicting the state of the atmosphere could be treated as an initial value problem of mathematical physics, wherein future weather is determined by integrating the governing partial differential equations, starting from the observed current weather. Today, this paradigm translates into solving a system of nonlinear differential equations at about half a billion points per time step and accounting for dynamic, thermodynamic, radiative, and chemical processes working on scales from hundreds of meters to thousands of kilometers and from seconds to weeks (Bauer et al. (2015)). The Navier–Stokes and mass continuity equations (including the effect of the Earth's rotation), together with the first law of thermodynamics

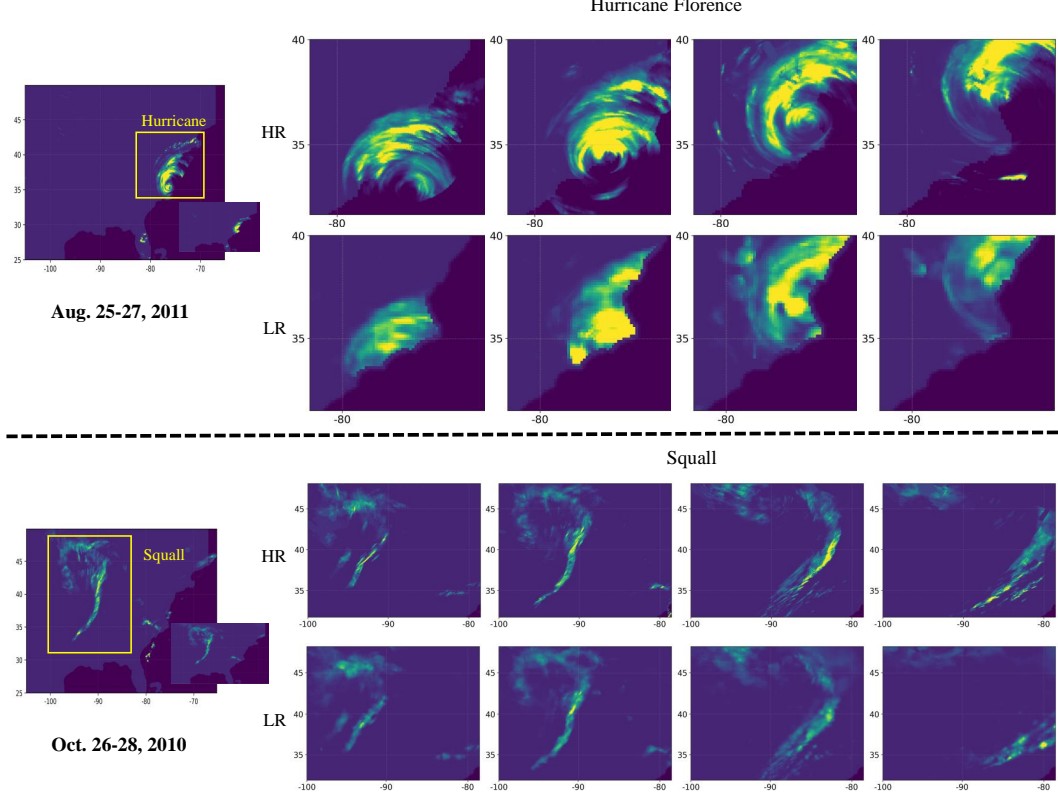

Figure 1: **Dataset Visualization**. Please zoom-in the figure for better observation. Please note that the details of the precipitation map are partially lost due to file compression. Here we plot 2 groups of typical meteorological phenomena (hurricane and squall) in the dataset. To learn more about the dataset, please visit our project website (coming soon) and supplementary material.

and the ideal gas law, represent the full set of prognostic equations in the atmosphere, describing the change in space and time of wind, pressure, density and temperature is described (formulas given in supplementary) (Bauer et al. (2015)). These equations have to be solved numerically using spatial and temporal discretization because of the mathematical intractability of obtaining analytical solutions, and this approximation creates a distinction between so-called resolved and unresolved scales of motion.

## 2.1 SPATIAL DOWNSCALING OF PRECIPITATION

The global weather forecast model, treated as a computational problem, relying on high-quality initial data input. The error of weather forecast would increase exponentially over time from this initial error of input dataset. Downscaling is one of the most important approaches to improve the initial input quality. Precipitation is one of the essential atmospheric variables that are related to daily life. It could easily be observed, by all means, *e.g.*, gauge station, radar, and satellites. Applying downscaling methods to precipitation and creating high-resolution rainfall is far more meaningful than deriving other variables, while it is the most proper initial task to test deep learning's power in geo-science. The traditional downscaling methods can be separated into dynamic and statistical downscaling.

Dynamic downscaling treats the downscaling as an optimization problem constraint on the physical laws. The dynamic downscaling methods find the most likely precipitation over space and time under the pre-defined physical law. It usually takes over 6 hours to downscale a 6-hour precipitation scenario globally on supercomputers (Courtier et al. (1994)). As the dynamic downscaling relying on pre-defined known macroscopic physics, a more flexible weather downscaling framework that

118 could easily blend different sources of observations and show the ability to describe more complex
119 physical phenomena on different scales is desperately in need.

120 Statistical downscaling is trying to speed up the dynamic downscaling process. The input of statisti-
121 cal downscaling is usually dynamic model results or two different observation datasets on different
122 scales. However, due to the quality of statistical downscaling results, people rarely apply statistical
123 downscaling to weather forecasts. These methods are currently applied in the tasks not requir-
124 ing high data quality but more qualitative understanding, *e.g.*, climate projection, which forecasts
125 the weather for hundreds of years on coarse grids and using statistical downscaling to get detailed
126 knowledge of medium-scale future climate system.

## 3 RAINNET: SPATIAL PRECIPITATION DOWNSCALING IMAGERY DATASET

### 3.1 DATA COLLECTION AND PROCESSING

129 To build up a standard *realistic (non-simulated)* downscaling dataset for computer vision, we
130 selected the eastern coast of the United States, which covers a large region (7 million $km^2$;
131 $105° \sim 65°W, 25° \sim 50°N$, GNU Free Documentation License 1.2) and has a 20-year high-quality
132 precipitation observations. We collected two precipitation data sources from National Stage IV QPE
133 Product (StageIV (Nelson et al. (2016)); high resolution at $0.04°$ (approximately $4km$), GNU Free
134 Documentation License 1.2) and North American Land Data Assimilation System (NLDAS (Xia
135 et al. (2012)); low resolution at $0.125°$ (approximately $13km$)). StageIV is mosaicked into a na-
136 tional product at National Centers for Environmental Prediction (NCEP), from the regional hourly/6-
137 hourly multi-sensor (radar+gauges) precipitation analyses (MPEs) produced by the 12 River Fore-
138 cast Centers over the continental United States with some manual quality control done at the River
139 Forecast Centers (RFCs). NLDAS is constructed quality-controlled, spatially-and-temporally con-
140 sistent datasets from the gauges and remote sensors to support modeling activities. Both products
141 are hourly updated and both available from 2002 to the current age.

142 In our dataset, we further selected the eastern coast region for rain season ($July \sim November$,
143 covering hurricane season; hurricanes pour over $10\%$ annual rainfall in less than 10 days). We
144 matched the coordinate system to the lat-lon system for both products and further labeled all the
145 hurricane periods happening in the last 17 years. These heavy rain events are the largest challenge
146 for weather forecasting and downscaling products. As heavy rain could stimulus a wide-spreading
147 flood, which threatening local lives and arousing public evacuation. If people underestimate the
148 rainfall, a potential flood would be underrated; while over-estimating the rainfall would lead to
149 unnecessary evacuation orders and flood protection, which is also costly.

### 3.2 DATASET STATISTICS

151 At the time of this work, we have collected and processed precipitation data for the rainy season
152 for 17 years from 2002 to 2018. One precipitation map pair per hour, $24$ precipitation map pairs
153 per day. In detail, we have collected $85$ months or $62424$ hours, totaling $62424$ pairs of high-
154 resolution and low-resolution precipitation maps. The size of the high-resolution precipitation map
155 is $624 \times 999$, and the size of the low-resolution is $208 \times 333$. Various meteorological phenomena
156 and precipitation conditions (*e.g.*, hurricanes, squall lines, *e.t.c.*) are covered in these data. The
157 precipitation map pairs in RainNet are stored in HDF5 files that make up 360 GB of disk space. We
158 select 2 typical meteorological phenomena and visualize them in Fig. 1. Our data is collected from
159 satellites, radars, gauge stations, *e.t.c.*, which covers the inherent working characteristics of different
160 meteorological measurement systems. Compared with traditional methods that generate data with
161 different resolutions through physical model simulation, our dataset is of great help for deep models
162 to learn real meteorological laws.

### 3.3 DATASET ANALYSIS

164 In order to help design a more appropriate and effective precipitation downscaling model, we have
165 explored the property of the dataset in depth. As mentioned above, our dataset is collected from mul-
166 tiple sensor sources (*e.g.*, satellite, weather radar, *e.t.c.*), which makes the data show a certain extent
167 of *misalignment*. Our efforts here are not able to vanquish the misalignment. This is an intrinsic

problem brought by the fusion of multi-sensor meteorological data. Limited by observation methods (*e.g.*, satellites can only collect data when they fly over the observation area), meteorological data is usually *temporal sparse*, *e.g.*, in our dataset, the sampling interval between two precipitation maps is one hour. The temporal sparse leads to serious difficulties in the utilization of precipitation sequences. Additionally, the movement of the precipitation position is directly related to the cloud. It is a fluid movement process that is completely different from the rigid body movement concerned in Super-Resolution. At the same time, the cloud will grow or dissipate in the process of flowing and even form new clouds, which further complicates the process. In the nutshell, although existed SR is a potential solution for downscaling, there is a big difference between the two. Especially, the three characteristics of downscaling mentioned above: *temporal misalignment*, *temporal sparse*, *fluid properties*, which make the dynamic estimation of precipitation more challenging.

## 4 EVALUATION METRICS

Due to the difference between downscaling and traditional figure super-resolution, the metrics that work well under SR tasks may not be sufficient for precipitation downscaling. By gathering the metrics from the meteorologic literature (the literature includes are Zhang & Yang (2004); Maraun et al. (2015); Ekström (2016); He et al. (2016); Pryor & Schoof (2020); Wootten et al. (2020)), we select and rename 6 most common metrics (a metrics may have multiple names in different literature) to reflect the downscaling quality: mesoscale peak precipitation error (MPPE), cumulative precipitation mean square error (CPMSE), heavy rain region error (HRRE) , cluster mean distance (CMD), heavy rain transition speed (HRTS) and average miss moving degree (AMMD).These 6 metrics can be separated as reconstruction metrics: MPPE, HRRE, CPMSE, AMMD, and dynamic metrics: HRTS and CMD.

The MPPE ($mm/hour$) is calculated as the difference of top quantile between the generated/real rainfall dataset which considering both spatial and temporal property of mesoscale meteorological systems, *e.g.*, hurricane, squall. This metric is used in most of these papers (for example Zhang & Yang (2004); Maraun et al. (2015); Ekström (2016); He et al. (2016); Pryor & Schoof (2020); Wootten et al. (2020) suggest the quantile analysis to evaluate the downscaling quality).

The CPMSE ($mm^2/hour^2$) measures the cumulative rainfall difference on each pixel over the time-axis of the test set, which shows the spatial reconstruction property. Similar metrics are used in Zhang & Yang (2004); Maraun et al. (2015); Wootten et al. (2020) calculated as the pixel level difference of monthly rainfall and used in He et al. (2016) as a pixel level difference of cumulative rainfall with different length of record.

The HRRE ($km^2$) measures the difference of heavy rain coverage on each time slide between generated and labeled test set, which shows the temporal reconstruction ability of the models. The AMMD ($radian$) measures the average angle difference between main rainfall clusters. Similar metrics are used in Zhang & Yang (2004); Maraun et al. (2015); Wootten et al. (2020) as rainfall coverage of a indefinite number precipitation level and used in He et al. (2016); Pryor & Schoof (2020) as a continuous spatial analysis.

As a single variable dataset, it is hard to evaluate the ability of different models to capture the precipitation dynamics when temporal information is not included (a multi-variable dataset may have wind speed, a typical variable representing dynamics, included). So here we introduce the first-order temporal and spatial variables to evaluate the dynamical property of downscaling results. Similar approaches are suggested in Maraun et al. (2015); Ekström (2016); Pryor & Schoof (2020). The CMD ($km$) physically compares the location difference of the main rainfall systems between the generated and labeled test set, which could be also understand as the RMSE of the first order derivative of precipitation data on spatial directions.The HRTS ($km/hour$) measures the difference between the main rainfall system moving speed between the generated and labeled test set which shows the ability for models to capture the dynamic property, which could be also understand as the RMSE of the first order derivative of precipitation data on temporal direction.Similar metrics are suggested in Maraun et al. (2015); Ekström (2016); Pryor & Schoof (2020) as the auto-regression analysis and the differential analysis.

More details about the metrics and their equations are given in supplementary materials. One metrics group (MPPE, HRRE, CPMSE, AMMD) mainly measures the rainfall deviation between the

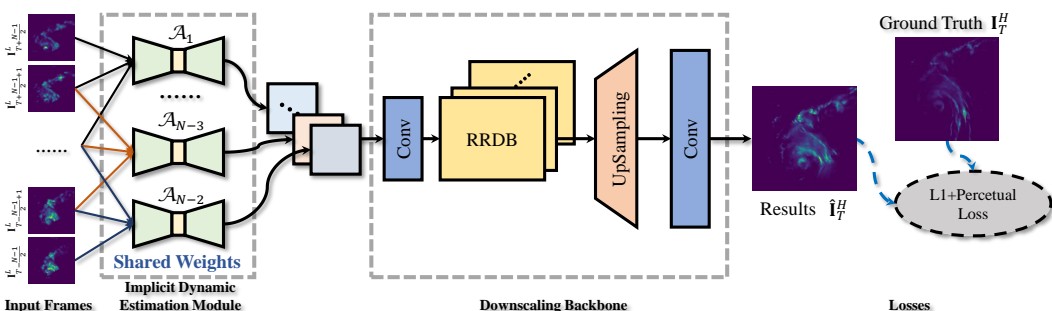

Figure 2: The pipeline of our proposed baseline model for spatial precipitation downscaling.

generated precipitation maps and GT. The other group (HRTS and CMD) mainly measures the dynamic deviation of generated precipitation maps. In order to further simplify the application of indices, we abstract them into two weighted and summed metrics: Precipitation Error Measure (PEM) and Precipitation Dynamics Error Measure (PDEM). We first align the dimensions of these two groups of metrics respectively. The first group of metrics (MPPE, HRRE, CPMSE, AMMD) is normalized, weighted and summed to get the precipitation error measure (PEM). According to Gupta et al. (1999), all the metrics are transferred to Percent Bias (PBIAS) to be suitable for metrics weighting. The original definition of PBIAS is the bias divided by observation, as $PBIAS = |Q_{model} - Q_{obs}|/|Q_{obs}|$. Here we rewrite the original metrics to PBIAS by dividing the metrics with annual mean observations of the original variables (AMO), as $PBIAS_i^{PEM} = |Metrics_i^{PEM}|/|AMO_i^{PEM}|, Metrics_i^{PEM} = \{MPPE, HRRE, CPMSE, AMMD\}$. In our dataset, $AMO_{MPPE}^{PEM} = 64$, $AMO_{HRREM}^{PEM} = 533$, $AMO_{CPMSE}^{PEM} = 0.64$, $AMO_{AMMD}^{PEM} = 332$, $AMO_{HRTS}^{PEM} = 15$, $AMO_{CMD}^{PEM} = 26$. The metrics then are ensembled to a single metric (PEM) with equal weight, as $PEM = \sum_i 0.25 \cdot PBIAS_i^{PEM}$. Following the same procedure, we then ensemble the second group of dynamic metrics (HRTS and CMD) to a single metrics $PDEM = \sum_i 0.5 \cdot PBIAS_i^{PDEM}$.

We also include the most common used metrics RMSE as one single metrics in our metrics list. RMSE could evaluate both reconstruction and dynamic property of the downscaling result.

## 5 APPLICATIONS OF RAINNET IN SPATIAL PRECIPITATION DOWNSCALING

As a potential solution, *Super-Resolution (SR)* frameworks are generally divided into the Single-Image Super-Resolution (SISR) and the Video Super-Resolution (VSR). Video Super-Resolution is able to leverage multi-frame information to restore images, which better matches the nature of downscaling. We will demonstrate this judgment in Sec. 6.1. The VSR pipeline usually contains three components: deblurring, inter-frame alignment, and super-resolution. Deblurring and inter-frame alignment are implemented by the motion estimation module. There are four motion estimation frameworks: 1). RNN based (Keys (1981); Tao et al. (2017); Huang et al. (2015); Haris et al. (2019)); 2). Optical Flow (Xue et al. (2019)); 3). Deformable Convolution based (Tian et al. (2020); Xiang et al. (2020); Wang et al. (2019)); 4). Temporal Concatenation (Jo et al. (2018); Caballero et al. (2017); Liao et al. (2015)). In fact, there is another motion estimation scheme proposed for the first time in the noise reduction task (Tassano et al. (2020)), which achieves an excellent video noise reduction performance. Inspired by (Tassano et al. (2020)), we design an implicit dynamics estimation model for the spatial precipitation downscaling. It is worth mentioning that our proposed model and the above four frameworks together form a relatively complete candidate set of dynamic estimation solutions.

**Proposed Framework.** As shown in Fig. 2, our framework consists of two components: *Implicit dynamic estimation module* and *downscaling Backbone*. These two parts are trained jointly. Suppose there are $N$ adjacent low-resolution precipitation maps $\{\mathbf{I}_{T-\frac{N-1}{2}}^L, .., \mathbf{I}_T^L, ..., \mathbf{I}_{T+\frac{N-1}{2}}^L\}$. The task is to reconstruct the high-resolution precipitation map $\mathbf{I}_T^H$ of $\mathbf{I}_T^L$. The implicit dynamic estimation module is composed of multiple vanilla networks $\mathcal{A} = \{\mathcal{A}_1, ..., \mathcal{A}_{N-2}\}$ ($N = 5$ in this paper) sharing weights. Each vanilla network receives three adjacent frames as input, outputs, and intermediate results. The intermediate result can be considered as a frame with implicit dynamic alignment. We

concatenate all the intermediate frames as the input of the next module. The specific structure of the vanilla network can be found in the supplementary materials. The main task of the downscaling backbone is to restore the high-resolution precipitation map $\mathbf{I}_T^H$ based on the aligned intermediate frames. In order to make full use of multi-scale information, we use multiple Residual-in-Residual Dense Blocks (Wang et al. (2018)) in the network. We employ the interpolation+convolution (Odena et al. (2016)) as the up-sampling operator to reduce the checkerboard artifacts. After processing by downscaling backbone we get the final estimated HR map $\hat{\mathbf{I}}_T^H$.

**Model objective.** The downscaling task is essentially to restore high-resolution precipitation maps. We learn from the super-resolution task and also apply $\mathcal{L}1$ and perceptual loss (Johnson et al. (2016)) as the training loss of our model. The model objective is shown below:

$$\mathcal{L}(\hat{\mathbf{I}}_T^H, \mathbf{I}_T^H) = \parallel \hat{\mathbf{I}}_T^H - \mathbf{I}_T^H \parallel_1 + \lambda \parallel \phi(\hat{\mathbf{I}}_T^H) - \phi(\mathbf{I}_T^H) \parallel_2, \tag{1}$$

where $\phi$ denotes the pre-trained VGG19 network (Simonyan & Zisserman (2015)), we select the $Relu5 - 4$ (without the activator (Wang et al. (2018))) as the output layer. $\lambda$ is the coefficient to balance the loss terms. $\lambda = 20$ in our framework.

## 6 EXPERIMENTAL EVALUATION

We conduct spatial precipitation downscaling experiments to illustrate the application of our proposed RainNet and evaluate the effectiveness of the benchmark downscaling frameworks. Following the mainstream evaluation protocol of DL/ML communities, cross-validation is employed. In detail, we divide the dataset into 17 parts (2002.7~2002.11, 2003.7~2003.11, 2004.7~2004.11, 2005.7~2005.11, 2006.7~2006.11, 2007.7~2007.11, 2008.7~2008.11, 2009.7~2009.11, 2010.7~2010.11, 2011.7~2011.11, 2012.7~2012.11, 2013.7~2013.11, 2014.7~2014.11, 2015.7~2015.11, 2016.7~2016.11, 2017.7~2017.11, 2018.7~2018.11) by year, and sequentially employ each year as the test set and the remaining 16 years as the training set, that is, 17-fold cross-validation. All models maintain the same training settings and hyperparameters during the training phase. These data cover various complicated precipitation situations such as hurricanes, squall lines, different levels of rain, and sunny days. It is sufficient to select the rainy season of the year as the test set from the perspective of meteorology, as the climate of one area is normally stable.

### 6.1 BASELINES

The SISR/VSR and the spatial precipitation downscaling are similar to some extent, so we argue that the SR models can be applied to the task as the benchmark models. The input of SISR is a single image, and the model infers a high-resolution image from it. Its main focus is to generate high-quality texture details to achieve pleasing visual effects. In contrast, VSR models input multiple frames of images (*e.g.*, 3 frames, 5 frames, *e.t.c.*). In our experiments, we employ 5 frames. The core idea of VSR models is to increase the resolution by complementing texture information between different frames. It is worth mentioning that VSR models generally are equipped with a motion estimation module to alleviate the challenge of object motion to inter-frame information registration.

We evaluated 7 state-of-the-art SISR frameworks (*i.e.*, Bicubic (Keys (1981)), SRCNN[1] (Dong et al. (2016)), SRGAN[2] (Ledig et al. (2017)),

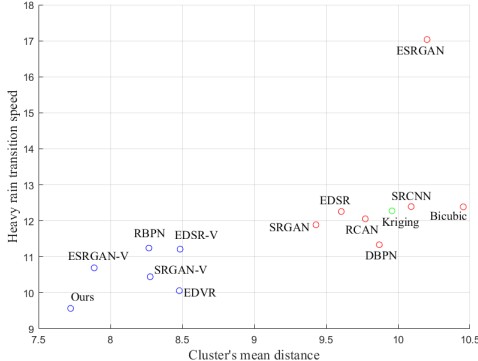

Figure 3: The dynamic property of benchmark algorithms. The frameworks of VSR are gathered in the lower-left corner of the figure, which demonstrates that VSR methods are superior to SISR and traditional methods in dynamic properties.

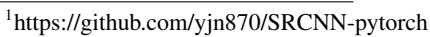

[1]https://github.com/yjn870/SRCNN-pytorch

[2]https://github.com/leftthomas/SRGAN

| Approach | MPPE↓ | HRRE↓ | AMMD↓ | CPMSE↓ | HRTS↓ | CMD↓ | PEM↓ | PDEM↓ | RMSE×100↓ |
|---|---|---|---|---|---|---|---|---|---|
| Kriging | 4.036 | 339.641 | 0.204 | 4.891 | 9.958 | 12.277 | 0.259 | 0.568 | 0.372 |
| Bicubic | 4.600 | 306.996 | 0.208 | 3.678 | 10.453 | 12.389 | 0.247 | 0.587 | 0.345 |
| SRCNN | 5.333 | 296.950 | 0.225 | 3.929 | 10.091 | 12.396 | 0.252 | 0.575 | 0.405 |
| SRGAN | 14.125 | 298.290 | 0.221 | 91.464 | 9.429 | 11.891 | 0.352 | 0.543 | 0.603 |
| EDSR | 4.748 | 288.354 | 0.204 | 3.292 | 9.605 | 12.259 | 0.236 | 0.556 | 0.329 |
| ESRGAN | 6.205 | 407.848 | 0.219 | 4.483 | 10.201 | 17.035 | 0.305 | 0.668 | 0.563 |
| DBPN | 6.596 | 302.278 | 0.212 | 5.692 | 9.869 | 11.336 | 0.256 | 0.547 | 0.380 |
| RCAN | 4.709 | 272.189 | 0.200 | 3.062 | 9.772 | 12.055 | 0.227 | 0.558 | 0.325 |
| SRGAN-V | 10.007 | 291.546 | 0.210 | 35.932 | 8.276 | 10.448 | 0.286 | 0.477 | 0.557 |
| EDSR-V | 4.592 | 289.331 | 0.201 | 3.269 | 8.484 | 11.214 | 0.235 | 0.498 | 0.323 |
| ESRGAN-V | 7.187 | 413.398 | 0.213 | 4.010 | 7.887 | 10.695 | 0.309 | 0.469 | 0.399 |
| RBPN | 4.816 | 287.214 | 0.201 | 2.680 | 8.267 | 11.244 | 0.235 | 0.492 | 0.317 |
| EDVR | 2.148 | 213.034 | 0.179 | 1.352 | 8.479 | 10.060 | 0.180 | 0.476 | 0.329 |
| Ours | 4.198 | 221.859 | 0.191 | 1.890 | 7.723 | 9.568 | 0.197 | 0.441 | 0.312 |

Table 1: Cross-validation results. Comparison with state-of-the-art super resolution approaches. The best performance is marked with red (1st best), blue (2nd best).

EDSR[3] (Lim et al. (2017)), ESRGAN[4] (Wang et al. (2018)), DBPN[5] (Haris et al. (2018)), RCAN[6] (Zhang et al. (2018)) and 5 VSR frameworks (*i.e.*, SRGAN-V, EDSR-V, ESRGAN-V, RBPN[7] (Haris et al. (2019)), EDVR[8] (Wang et al. (2019)), of which 3 VSR methods (*i.e.*, SRGAN-V, EDSR-V, ESRGAN-V) are modified from SISR. In particular, we build SRGAN-V, EDSR-V and ESRGAN-V by concatenating multiple frames of precipitation maps as the input of the model. In addition, we also evaluated the traditional statistics method Kriging (Stein (2012)), which is widely applied in weather forecasting. The mentioned 8 metrics are used to quantitatively evaluate the performance of these SR models and our method. Further, we select some disastrous weather as samples for qualitative analysis to test the model's ability to learn the dynamic properties of the weather system. And we employ the implementation of Pytorch for Bicubic. We use 4 NVIDIA 2080 Ti GPUs for training. We train all models with following setting. The batch size is set as 24. Precipitation maps are random crop into $64 \times 64$. We employ the Adam optimizer, beta1 is 0.9, and beta2 is 0.99. The initial learning rate is 0.001, which is reduced to 1/10 every 50 epochs, and a total of 200 epochs are trained. We evaluate benchmark frameworks with 17-fold cross-validation. The downscaling performances are shown in Tab. 1. We divide the indicators mentioned above into two groups. PDEM measures the model's ability to learn the dynamics of precipitation. PEM illustrates the model's ability to reconstruct precipitation.

From Tab. 1, we can learn that the overall performance of the VSR methods are better than SISR models, which shows that the dynamic properties mentioned above are extremely important for the downscaling model. Furthermore, it can be seen from Fig. 3 that the SISR method is clustered in the upper right corner of the scatter plot, and the VSR method is concentrated in the lower-left corner, which further shows that the dynamic properties of the VSR methods are overall better than the SISR methods. In addition, our method achieves the $1st$ best performance in RMSE, PDE, and achieve the second-best performance on PEM. The score shows that the implicit dynamic estimation framework

---

[3]https://github.com/sanghyun-son/EDSR-PyTorch

[4]https://github.com/xinntao/ESRGAN

[5]https://github.com/alterzero/DBPN-Pytorch

[6]https://github.com/yulunzhang/RCAN

[7]https://github.com/alterzero/RBPN-PyTorch

[8]https://github.com/xinntao/EDVR

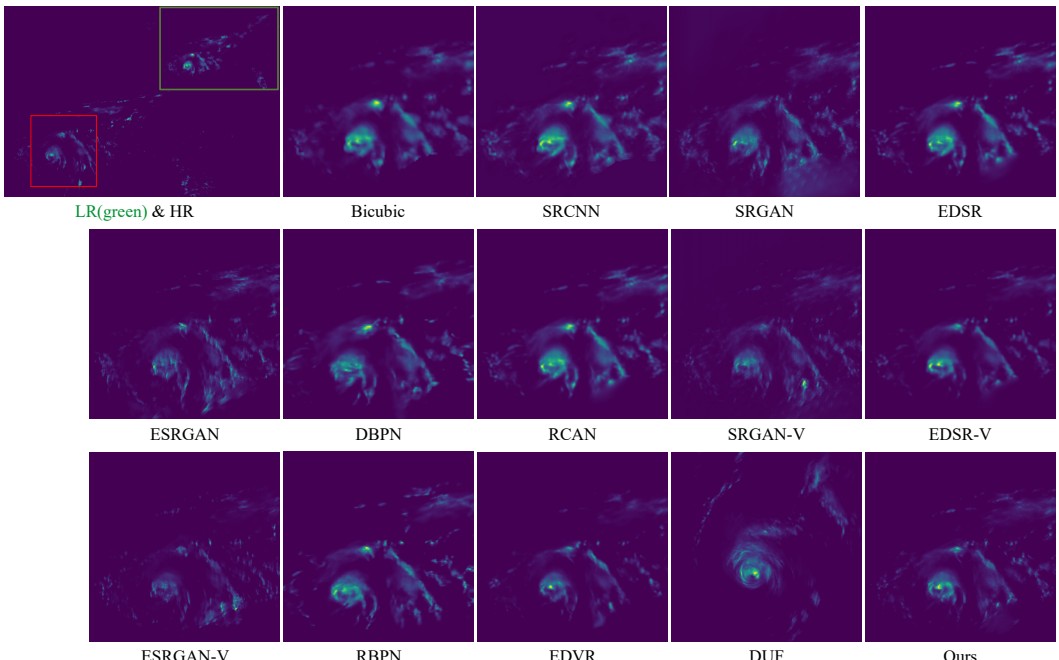

Figure 4: Visual comparison with state-of-the-art Super Resolution approaches. Please zoom-in the figure for better observation. More results can be found in suppl.

used is feasible and effective. It is worth mentioning that the traditional downscaling method Kriging performs better than many deep learning models (*e.g.*, SRGAN, ESRGAN)

### 6.1.1 QUALITATIVE ANALYSIS

We visualized the tropical cyclone precipitation map of the $166th$ hour (6th) in September 2010 and the high-resolution precipitation map generated by different methods. As shown in Fig. 4, the best perceptual effects are generated by EDVR and Our framework. Zooming in the result image, the precipitation maps generated by SRGAN and EDSR present obvious checkerboard artifacts. The reason for the checkerboard artifacts should be the relatively simple and sparse texture pattern in precipitation maps. The results generated by Bicubic, RCAN, Kriging, and SRCNN are over-smooth. DBPN even cannot reconstruct the eye of the hurricane. Especially, the result generated by Kriging is as fuzzy as the input LR precipitation map. In conclusion, the visual effects generated by the VSR methods are generally better than the SISR methods and the traditional method. From the perspective of quantitative and qualitative analysis, the dynamics estimation framework is very critical for downscaling.

## 7 CONCLUSION

In this paper, we built the first large-scale *real* precipitation downscaling dataset for the deep learning community. This dataset has 62424 pairs of HR and LR precipitation maps in total. We believe this dataset will further accelerate the research on precipitation downscaling. Furthermore, we analyze the problem in-depth and put forward three key challenges: temporal misalignment, temporal sparse, fluid properties. In addition, we propose an implicit dynamic estimation model to alleviate the above challenges. At the same time, we evaluated the mainstream SISR and VSR models and found that none of these models can solve RainNet's problems well. Therefore, the downscaling task on this dataset is still very challenging.

This work still remains several open problems. Currently, the data domain of this research is limited to the eastern U.S. In future research, we would enlarge the dataset to a larger domain. The dataset is only a single variable now. In future research, we may include more variables, e.g. temperature and wind speed.

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
