# OpenReview forum: "RainNet: A Large-Scale Imagery Dataset for Spatial Precipitation Downscaling"
_ICLR.cc/2022/Conference — ICLR 2022 Submitted_

### Official Review · Reviewer_ggKX · 2021-10-23

**Correctness:** 3
**Technical Novelty And Significance:** 3
**Empirical Novelty And Significance:** 3
**Recommendation:** 6
**Confidence:** 4

**Main Review:**

Strength:
- Creates and releases a high/low res pair dataset of real images which are ML-ready and instrumental in kicking of climate research.
- Provide well thought out experiments (reproducing other methods etc) along with their baselines on their new metric

Weakness:
- One way to evaluate how much better are high res images is to use them for potential Applications to downstream tasks which is not provided.
- Any method that converts low res to high res, generates some information and so uncertainty maps help instill confidence in predictions. The paper does not provide any uncertainty estimation.
- Qualitative analysis of methods such as visually seeing their utility could help a domain scientist use this method; but qualitative analysis is lacking.

**Summary Of The Paper:**

The paper provides a baseline method that generates a dataset of high+low res precipitation maps. The paper argues that precision of weather event predictions depend on the imagery resolution and hence having high res maps will lead to better weather prediction.  The eventual dataset that is formed contains a variety of diverse climate events like hurricanes etc and the pair of high/low res forms the necessary annotation.

**Summary Of The Review:**

Overall well written. As the authors mention Ml-readiness; its also important to think about how will an actual domain scientist use this work. This is where evaluating downstream applications and understanding uncertainty in the results is crucial. This part is missing, other than that its a clear and concise approach.

---

### Official Review · Reviewer_5pVg · 2021-10-25

**Correctness:** 2
**Technical Novelty And Significance:** 3
**Empirical Novelty And Significance:** 2
**Recommendation:** 3
**Confidence:** 5

**Main Review:**

The topic of the paper is relevant -- a large scale dataset of gridded, paired, high-resolution and low-resolution precipitation measurements would be both interesting and highly useful for exploring deep learning approaches to for earth science applications. However, this paper is lacking in several key dimensions:

- First, the paper does not adequately describe the proposed dataset itself:
  - From Section 3.1 it seems that the data comes from the NLDAS dataset and the NCEP Stage IV 4km gridded precipitation data. More details are needed here, for example, the NLDAS dataset has two versions (001 having been discontinued in April 2020). Which products exactly were used? What are the limitations of these datasets (e.g. NLDAS is a reanalysis dataset)?
  - The resolution of the high-resolution data is 0.04 degrees and the resolution of the low-resolution data is 0.125 degrees, however the high-resolution and low resolution images are exactly 3x different, which implies that a resampling step was used at some point and should be detailed.
  Why were image sizes of 624x999 and 208x333 used? Practically, in most deep learning cases these will need to be cropped/resized anyway to accommodate downsampling.
  - Figure 1 seems to show that the low resolution data is clipped to the coastline? If this is the case, then it would present a serious limitation of the dataset as the high-resolution data looks like it is _not_ clipped to the coast.
  - The paper argues that this is the first _real_ dataset for studying the downscaling task, but it does not argue why it is insufficient to simply downsample the high-resolution data to get low-resolution data. If it is the case that there is a large difference between the downsampled high-resolution data and the low-resolution data, then this should be discussed.
- Second, the reported results (Table 1) are from a 17 fold cross-validation procedure over time, however are shown without standard deviations or confidence intervals. The difference between some methods is very small, and could easily be insignificant. Deep learning approaches can have large differences in performance with everything but random number seed used in training held the same, therefore it is important to at least report standard deviation over runs. As is, it is difficult to draw conclusions about the quality of the different approaches.
- Finally, there are many claims and details throughout the paper that need to be cited/addressed. For example:
  - The first sentence of Section 2 is incorrect. Bauer et al. (2015) explain that the beginning of the 20th century (not the beginning of the 19th century as written in line 89) is when "geoscientists recognized that predicting the state of the atmosphere could be treated as an initial value problem...". The Navier-Stokes equations were only discovered in the mid-19th century.
  - (Line 252) "It is worth mentioning that our proposed model and the above four frameworks together form a relatively complete candidate set of dynamic estimation solutions." -- this is a strong, unnecessary, claim without any support -- how do the proposed model and four frameworks form a relatively complete set of dynamic estimation solutions?
  - Lines 131 and 133 includes "GNU Free Documentation License 1.2" completely out of context?
  - "The precision of weather and climate prediction is highly dependent on the resolution and reliability of the initial environmental input variables, and spatial precipitation downscaling is the most promising solution." -- this is a bold claim and needs a citation.
  - "The error of weather forecast would increase exponentially over time from this initial error of input dataset." (lines 105-106) the error _does_ increase exponentially over time.

Minor points:
- "e.t.c." --> "etc." throughout
- "Unfortunately, there are looming issues hinders the" --> "Unfortunately, there are looming issues that hinder the"
- "Rainnet" --> "RainNet" throughout
- "quantile-quantile maps to analyzing the maps." (lines 56-57) doesn't make sense
- "Hurricane, Squall" --> "hurricane, squall" (line 59)
- "alleviate these obstacles" consider rewording "alleviate these challenges", "overcome these obstacles"
- "REAL" --> "real" (line 82)
- Sentence incomplete (line 104-105)
- Caption of Figure 1 says, "To learn more about the dataset, please visit our project website (coming soon) and supplementary material." The main contribution of the paper, however, is this dataset, therefore the caption should reference the paper.
- Line 219 says more details about the metrics is given in the supplementary materials, however the supplementary materials first copies the text from the main paper word-for-word (including the sentence about referencing the supplementary material...)
- The list of dates in line 279-282 should be described differently.
- In the Proposed Framework subsection (line 255) H, L, and T are not described.
- Consider using font formatting instead of coloring in Table 1 to make the table more easily interpreted by readers with colorblindness. The red highlighting, in particular, will be difficult for readers with the most common form of colorblindness.
- Figure 4, caption is potentially unfinished.

**Summary Of The Paper:**

This paper proposes a dataset called RainNet for studying precipitation downscaling. RainNet consists of ~62k pairs of low-res/high-res precipitation maps from various meteorological events over the east coast of the US over 17 years. This dataset was created from two existing products: NLDAS and the NCEP's Stage IV 4km gridded precipitation data. Further, the paper proposes two metrics, Precipitation Error Measure (PEM) and Precipitation Dynamics Error Measure (PDEM). Finally, the paper compares several superresolution methods from computer vision literature, traditional statistical approaches to downscaling, as well as the proposed architecture/method on the proposed dataset to form a set of benchmark results.

**Summary Of The Review:**

This paper should be rejected. Its main contribution is the dataset, RainNet, and benchmark results on RainNet. However the paper does not adequately describe the dataset, and the benchmark results are not reported with measures of uncertainty, which makes comparisons impossible.

---

### Official Review · Reviewer_szBD · 2021-11-02

**Correctness:** 4
**Technical Novelty And Significance:** 3
**Empirical Novelty And Significance:** Not applicable
**Recommendation:** 5
**Confidence:** 4

**Main Review:**

There is undoubtedly a need for large labeled and trustworthy datasets for segmentation, detection, tracking of large-scale weather events including extreme weather events. From that perspective, this paper is a good contribution to the field of AI-driven meteorology where such datasets are in abundant need. The evaluation of the different models on this dataset provides a good benchmark as well. This is well written and deserves to be published although I'm unsure whether it should be in ICLR. I feel that a more meteorology-focused journal is a better venue for this work.

**Summary Of The Paper:**

This paper presents a $0.4^\circ$ dataset for precipitation downscaling. It further shows 14 SOTA models with/without ML for evaluation on the dataset.

**Summary Of The Review:**

As I mentioned in my review, this is a great paper proposing a new dataset that is well constructed and evaluated and is high-resolution. However, I don't feel that ICLR is the right venue for this work.

---

### Official Review · Reviewer_D3tQ · 2021-11-07

**Correctness:** 3
**Technical Novelty And Significance:** 2
**Empirical Novelty And Significance:** 3
**Recommendation:** 3
**Confidence:** 5

**Details Of Ethics Concerns:**

It is very disappointing that the authors chose to resubmit a paper without fully addressing the feedback provided by reviewers in the previous submission. While the authors did make some changes to the text and experimental results to address some issues, the failure to address even very basic issues like typos and grammatical errors makes this resubmission look more like fishing for different reviewers than a genuine effort to improve the paper. This appears to me to be somewhat ethically questionable, but I would appreciate an ethics review to get a more objective opinion.

**Main Review:**

This work was recently submitted to NeurIPS 2021. ~~I would highly recommend that all reviewers and area chairs read the discussion there and compare the previously submitted version of this work to the version that is under review here.~~

EDIT: Unfortunately, NeurIPS changed their visibility policy in 2021 for rejected submissions, so the discussion is no longer available to the public. Apologies for the confusion.

The authors have made a few improvements to address the concerns raised in this original discussion:

1) Cross validation results are presented to provide a more robust comparison between models.
2) The discussion of metrics in section 4 is much more thorough than in the previous version.

Unfortunately, several issues remain unaddressed.

1) **Quantitative results do not include standard errors or any indication of the variance in cross validation.** This was one of the top concerns raised in the previous discussion. The per-fold results posted by the authors in this thread show a substantial amount of variance in the relative ranking of the evaluated methods between folds, so this is very important for transparency and fair model comparison.
2) **Inadequate discussion of hyperparameter tuning, particularly for the baselines.** The description of hyperparameter settings improved in this version of the work, but the discussion on how hyperparameters were selected did not. There is no discussion of hyperparameter tuning methodology and/or how the hyperparameters for baseline methods were chosen. It is both unfair and uninteresting to compare a model which was tuned for a statistical downscaling task to a baseline which was tuned for natural image super resolution.
3) **Inadequate framing of work in context of existing literature.** The literature review is sparse and does not provide any discussion of how the authors work fits into the broader context of the literature on applying deep learning to statistical downscaling. There is also no discussion of existing frameworks often used for downscaling evaluation such as VALUE and Climdex (citations are provided but there are no mentions in the text).
4) **Poor presentation of qualitative results.** Figure 4 in particular is difficult to evaluate due to the poor rendering quality and presentation format. As mentioned in the previous discussion, the low/high resolution ground truth are too small relative to the other images for meaningful visual comparison.
5) **Language and formatting errors.** The text is still littered with typos, grammatical errors, and problematic (sometimes even confusing) diction. Several such issues that were pointed out directly by reviewers in the previous discussion still remain, which seems to indicate that the authors did not bother to further proof-read or revise the text.

**Summary Of The Paper:**

The authors present a novel, high resolution dataset for precipitation downscaling collected from observational data in the southeastern US. In addition to the dataset, several novel metrics for evaluating statistical downscaling methods are presented, as well as a novel deep learning algorithm based on video super resolution and "implicit dynamics" estimation.

**Summary Of The Review:**

The dataset and metrics presented by the authors are valuable contributions, but the paper has significant problems with framing, presentation, quantitative evaluation, and language/formatting. Due to these quality deficiencies, I cannot recommend acceptance of this paper in its current form.

---

### Decision · Program_Chairs · 2022-01-20

**Decision:**

Reject

**Comment:**

This paper proposes a new dataset, called RainNet, obtained from gridded precipitation data, for training precipitation downscaling methods, as well as a new neural network-based architecture for that task, which estimates the underlying dynamics of the local weather system, and new metrics for evaluating precipitation downscaling methods.

Reviewers praised the large, novel and useful dataset (D3tQ, szBD, ggKX) and novel metrics for evaluating statistical downscaling methods (D3tQ), along with evaluation on 14 baselines (szBD, ggKX).

There were however many issues highlighted by the reviewers. First, reviewer D3tQ raised concerns about the paper being resubmitted after rejection from NeurIPS (/pdf?id=VVZZJiQB51l), with minimal changes (/pdf?id=6p8D4V_Wmyp), and noticed that the authors did not follow up on most reviewer recommendations. D3tQ noticed however that in the ICLR resubmission, the cross validation results were presented to provide a more robust comparison between models, and that the discussion of metrics in section 4 was much more thorough than in the previous version.

Other themes in the negative reviews included concerns about missing standard errors in the cross-validation results (D3tQ, 5pVg) or measures of uncertainty in the upscaling (ggKX), lack of information about hyperparameter tuning (D3tQ), inadequate literature review about statistical downscaling (D3tQ), lack of information about the dataset (5pVg), missing discussion about applications (ggKX) and insufficient proofreading (D3tQ, 5pVg).

I will not take into consideration the criticism from szBD who "don't feel that ICLR is the right venue for this work" as I do not find such opinions to be much helpful.

The authors did not provide a rebuttal to the initial reviews and there was no discussion about this paper among the reviewers. Given the issues raised by the reviewers and the scores of 3, 3, 5 and 6, I believe that this paper does not meet the acceptance bar in its current form.

Sincerely,
AC